# Emerging Therapeutic and Inflammation Biomarkers: The Role of Meteorin-Like (Metrnl) and Follistatin-Like 1 (FSTL1) in Inflammatory Diseases

**DOI:** 10.3390/ijms26199711

**Published:** 2025-10-06

**Authors:** Tsvetelina Velikova, Konstantina Bakopoulou, Milena Gulinac, Evelina Manova, Hristo Valkov, Dimitrina Miteva, Russka Shumnalieva

**Affiliations:** 1Medical Faculty, Sofia University St. Kliment Ohridski, 1 Kozyak Street, 1407 Sofia, Bulgaria; mgulinac@hotmail.com (M.G.); d.georgieva@biofac.uni-sofia.bg (D.M.); rshumnalieva@yahoo.com (R.S.); 2Faculty of Medicine, Medical University Sofia, Blvd. ‘Akademik Ivan Evstratiev Geshov’ 15, 1431 Sofia, Bulgaria; bakopouloukonstantina@gmail.com; 3Department of General and Clinical Pathology, Medical University of Plovdiv, Bul. Vasil Aprilov 15A, 4000 Plovdiv, Bulgaria; 4Department of Genetics, Faculty of Biology, Sofia University “St. Kliment Ohridski”, 8 Dragan Tzankov Str., 1164 Sofia, Bulgaria; emariova@uni-sofia.bg; 5Clinic of Gastroenterology, University Hospital “Tsaritsa Yoanna-ISUL”, “Byalo More” No. 8, 1527 Sofia, Bulgaria; hristo.valkov.m.d@gmail.com; 6Department of Gastroenterology, Faculty of Medicine, Medical University of Sofia, Blvd. ‘Akademik Ivan Evstratiev Geshov’ 15, 1431 Sofia, Bulgaria; 7Clinic of Rheumatology, University Hospital St. Anna, Blvd. Dimitar Mollov, 1709 Sofia, Bulgaria; 8Department of Rheumatology, Faculty of Medicine, Medical University of Sofia, Urvich Street 13, 1612 Sofia, Bulgaria

**Keywords:** biomarkers, Metrnl, FSTL1, inflammation, autoimmune rheumatic diseases, gastrointestinal inflammatory diseases, metabolic diseases

## Abstract

In recent years, Meteorin-like protein (Metrnl/IL-41) and Follistatin-like 1 (FSTL1) have emerged as multifunctional molecules that play roles in immunity, metabolism and tissue remodeling. Although they demonstrate pleiotropic effects, they are promising candidates for biomarkers and possible therapeutic targets. The development of new, disease-specific biomarkers will enable clinicians to more effectively monitor inflammatory activity, more accurately assess disease severity, better predict survival, and select appropriate medical treatments. In this review, we present the role of Meteorin-Like Protein (Metrnl) and Follistatin-like 1 (FSTL1) in inflammation in autoimmune rheumatic diseases, as well as in other autoimmune pathologies, cardiovascular diseases, and metabolic diseases. Metrnl, widely expressed in different tissues and organs, is very important for inflammation, immune responses and metabolic disorders. FSTL1 also shows dynamic changes in its expression through various diseases, including cardiovascular conditions, cancer, asthma, and arthritis. Both proteins participate in multiple important signaling pathways, and understanding their diagnostic and therapeutic potential holds great scientific interest. Their complex nature requires careful evaluation of safety concerns and translation to clinical practice.

## 1. Introduction

Inflammation is defined as a local response or defense reaction of living tissue to injury by any agent to eliminate or limit the spread of such an agent, as well as removing dead tissue and necrotic cells [1]. Depending on the host’s defense capacity, the inflammatory response can be classified as acute and chronic. In addition, factors that determine variations in the inflammatory response by the host include the general health condition of the host, the presence of chronic conditions, immune status (immunodeficiency), and the type or site of the affected tissue. During inflammation, small blood vessels in the body undergo significant changes. They include transient vasoconstriction of arterioles, persistent progressive vasodilation, an increase in local hydrostatic pressure, and stasis of blood flow, followed by leukocyte margination and/or emigration [1].

Regardless of whether the etiology of inflammation is infectious (bacteria and/or viruses) or non-infectious (physical, chemical and/or biological factors), the body reacts to the damage by sending out special chemical signals. It stimulates multiple signaling pathways that activate the production of inflammatory mediators and promote the healing of affected tissues.

Inflammatory mediators are key biochemical agents that play a central role in the body’s inflammatory response. The sources of these mediators include plasma and various cells, such as platelets, basophilic and eosinophilic granulocytes, lymphocytes, and macrophages. Their activation initiates a complex cascade of events driven by proinflammatory cytokines [2]. Cytokines are critical to both the acute and, more prominently, the chronic phases of inflammation. Among the principal cytokine mediators are interleukin-1 (IL-1), tumor necrosis factor-alpha (TNF-α), tumor necrosis factor-beta (TNF-β), interferon-gamma (IFN-γ), and chemokines. It is essential to recall that activated T-lymphocytes produce TNF-β and IFN-γ, while macrophages primarily secrete IL-1 and TNF-α. Key inflammatory mediators, such as IL-1, TNF-α, and TNF-β, contribute to various immunological processes, including enhanced leukocyte adhesion, increased cytokine production, and the initiation of the acute-phase response. IFN-γ, in particular, causes activation of macrophages and neutrophil leukocytes. TNF-α, IL-1, and IFN-γ are also essential for promoting the synthesis of other proinflammatory cytokines, such as IL-6 and IL-8, and for initiating the production of lipid mediators like prostaglandins and leukotrienes [2].

In response to infectious agents or tissue damage, IL-6 is rapidly produced, contributing to host defense and participating in various immune responses. Although its expression is tightly regulated transcriptionally and post-transcriptionally, its continuous synthesis can lead to negative effects on inflammation and autoimmune diseases (e.g., rheumatoid arthritis, RA) [3,4]. It can also bind to other cells through cytokine receptor type I, activating various transcription factors such as the Janus kinase/signal transducer and activator of transcription (JAK/STAT) pathway, thereby enhancing the proinflammatory response. IL-6 is considered one of the main factors for the secretion of most acute-phase proteins (APPs) from the liver. Therefore, IL-6 serves as a mediator with pleiotropic effects on inflammation, liver regeneration, and the immune response.

Currently, two main groups of inflammatory biomarkers are used in clinical practice, including first-line circulating cytokines such as interleukins (IL) and interferons (IFNs). Menzel et al. have described in detail why some mediators are not used in routine practice, with the main reasons being that they are too expensive and difficult to store [2]. Other mediators are also suitable and have shown a good association with the clinical outcome of the disease. These include acute-phase proteins (APPs), which can inhibit microbial growth, are formed in the early stages of inflammation, and are secreted by the liver. Such examples include C-reactive protein (CRP), haptoglobin, or serum amyloid A (SAA) [2].

Moreover, it is essential to note that, like other serum inflammatory markers, CRP has a relatively brief half-life; therefore, proper sample handling is crucial. Simultaneously, conditions such as a cold or small tissue injury can markedly affect individual serum CRP levels. Chronic diseases, including type 2 diabetes mellitus (T2DM), chronic heart failure, and obesity, can influence CRP levels, thereby complicating subsequent analyses [5].

Recently, we have all been faced with a “new” disease— Coronavirus disease 2019 (COVID-19). Still, our understanding of the disease is no longer limited, and we can now determine the course and outcome of COVID-19 by monitoring laboratory inflammatory biomarkers. Many of the patients in whom the disease was very severe were those in whom the so-called cytokine storm occurred. This raised the question of whether it is necessary to study new inflammatory biomarkers that would find wider application in clinical practice. Several new and different biomarkers have been rediscovered so far, which, in combination with the already known ones, improve the prognostic prediction of severity and/or mortality from COVID-19. Galliera et al. described in a longitudinal study a new infectious biomarker— soluble CD14 subtype (SCD14-ST), which was significantly higher in patients who eventually died [6]. Literature data show that organokines are potential biomarkers of inflammatory disease severity. Their impaired profiles are frequently connected with endothelial dysfunction, cytokine storm, and multiple organ failure in COVID-19 patients. Recent studies suggest a potential relationship between organokines, such as Meteorin-like protein (Metrnl), and the pathophysiology of COVID-19 [7,8]. Their findings are initial but underscore the important role of Metrnl in immune dysregulation during viral infections.

In another study, Zhan et al. demonstrated that patients with elevated levels of endocan, galectin-3, sCD14-ST, soluble urokinase plasminogen activator receptor (suPAR), and monocyte distribution width (MDW) tended to develop worse overall clinical outcomes of the disease [9]. A recently published study also described the importance of new inflammatory biomarkers and their relevance again in patients with COVID-19 [10]. Data indicated that serum levels of Pentraxin-3 (PTX3) and soluble urokinase Plasminogen Activator Receptor (suPAR) were markedly enhanced in 150 patients, except for those in the intensive care unit, whose levels were diminished. The analysis also included data on leukocytes, neutrophils, neutrophil/lymphocyte ratio (NLR), troponin, procalcitonin (PCT), D-dimer, CRP, lymphocytes, and ferritin.

All of these studies support the initial hypothesis that the presence of novel biomarkers, such as those described above, combined with conventional inflammatory biomarkers, can more accurately predict the outcome of COVID-19 disease.

## 2. Meteorin-like Protein (Metrnl) as a Novel Inflammatory Biomarker in Autoimmune Rheumatic Diseases

### 2.1. Structure and Biological Function of Metrnl

Meteorin-like protein (Metrnl), also known as IL-41, subfatin, and cometin, is a pleiotropic secretory protein. It is composed of 311 amino acids, its gene is located on chromosome 17 (17q25.3), and it is involved in many biological processes [11]. Its expression has been identified in a range of mesenchymal-derived cell types, including osteoblasts, macrophages, and smooth muscle cells, as well as in granulocytes and dendritic cells. Notably, Metrnl is not produced by cells of lymphoid origin [12]. As the name “subfatin” suggests, large amounts of it are found in subcutaneous white adipose tissue in animals and humans, as well as in perivascular, interscapular, greater omentum, and subcutaneous fat [13]. Metrnl was originally described as a homologue of meteorin, a neurotrophic factor that is highly expressed in the central nervous system (CNS) [12,14]. Conversely, small amounts of it are found in brown adipose tissue [13,14]. However, Metrnl levels in cerebrospinal fluid (CSF) appear to be highly dependent on the integrity of the blood–brain barrier (BBB) [15]. Other sites of high Metrnl expression include the liver, muscle, spleen, heart, brain, and thymus [12]. Furthermore, Metrnl is highly induced in barrier tissues, including skin and mucosa, by resting fibroblasts and keratinocytes, as well as alternatively activated macrophages, suggesting its active role in innate immunity [11,13]. The biological profile of Metrnl appears to be quite complex and heterogeneous, encompassing a variety of anti-inflammatory, immunomodulatory, metabolic, osteoregulatory, and neuroregulatory effects [11,13].

### 2.2. Metrnl in Immune Regulation and Inflammation

Metrnl expression in macrophage cells, assessed using enzyme-Linked Immunosorbent Assay (ELISA) by Uschach et al., was shown to be elevated in response to M2 polarization induced by IL-4 and IL-13, while M1-polarisation led to its downregulated expression [11]. Bone marrow-derived macrophages can stimulate Metrnl expression via the release of TNF-α, IL-4, IL-12, and IL-17α, while its downregulation is dependent on the action of Transforming growth factor-beta (TGF-β) and IFN-γ. Metrnl-treated macrophages demonstrated increased levels of inflammatory cytokines, including IL-6, IL-10, C-X-C motif ligand 1 (CXCL1), and C-C motif ligand 2 (CCL2) [16].

The TLR4 agonist lipopolysaccharide (LPS) can also stimulate Metrnl expression in murine macrophages. This leads to in vivo inflammatory responses through the release of different cytokines [11,14]. Its high expression in the digestive, respiratory, and integumentary systems, all of which serve as entry points for foreign pathogens, may explain why it has been implicated in innate immunity. Metrnl is abundantly expressed in skin fibroblasts, with increased levels observed in both psoriasis and the synovial membrane of individuals with RA [14].

Metrnl can modulate the immune system. Knockout mice demonstrate animals show reduced serum antibody titers, which is associated with compromised immunological function, reduced serum antibody levels, increased susceptibility to sepsis, spontaneous immune responses, and endotoxin shock [13,17]. A hyporesponsive T helper cell type 2 (Th2) phenotype is observed with genetic alterations in the gene, indicating that this cytokine is crucial for maintaining balanced Th2-dependent immune responses [16,18]. Metrnl also ameliorates lipid-induced inflammation/insulin resistance via AMP-activated protein kinase/Peroxisome proliferator-activated receptor delta (AMPK/PPARδ) in muscle [19]. These data demonstrate that Metrnl is a key regulator of immune competence, linking innate Th2 pathways to adaptive humoral immunity.

A key feature of Metrnl is its anti-inflammatory activity. This is demonstrated by its capacity to diminish vascular inflammation by limiting the production of reactive oxygen species (ROS) and reducing the recruitment of the NLR family pyrin domain containing 3 (NLRP3) inflammasome component [14,20]. Metrnl most likely exerts its action through the activation of AMP-kinase (AMPK), which downregulates the mammalian target of rapamycin (mTOR) [14]. It is worth noting that the mTOR pathway has multiple known associations with tumor progression and angiogenesis, which justifies the correlation of Metrnl levels with various malignancies [21]. The mTOR signaling pathway controls and regulates cellular growth, metabolic processes, and angiogenesis. Its abnormal function leads to cancer development and progression. Metrnl activates AMPK and reduces mTOR activity. Studies show that modifying Metrnl expression or activity affects cancer survival rates and disease progression in multiple types of cancer [16,22]. Wang et al. identified Metrnl as a candidate biomarker and therapeutic target in various conditions, including cancer, highlighting its pleiotropic role [23]. The data support the association between Metrnl levels and cancer development and require context-specific evaluation.

Indeed, protumorigenic and antiapoptotic effects of Metrnl have been shown by Manning et al. in the context of pancreatic cancer [24] and by Xiao et al. for bladder cancer [25]. The potent angiogenic effect attributed to Metrnl is further elucidated by the failure of Metrnl knockout mice to mount a KIT-dependent post-infarction angiogenic response in ischemic heart injury [26,27]. In addition to its action via AMPK phosphorylation, Metrnl also upregulates PPARδ expression, as demonstrated in LPS-treated human umbilical vein endothelial cells (HUVECs) and human THP-1 monocytes. Furthermore, a general inhibitory effect on apoptosis and adhesion molecule production was observed, further supporting the beneficial impact of Metrnl against endothelial inflammation [14,28].

Convincing data on Metrnl’s anti-inflammatory action were earlier demonstrated in H9C2 cells that underwent oxygen and glucose deprivation followed by reperfusion. In these cells, Metrnl overexpression reduced apoptosis, inflammation, and endoplasmic reticulum stress induced by myocardial reperfusion injury via activation of the AMPK-PAK2 (serine/threonine protein kinase) signaling pathway [29]. Freedman et al. postulated that Metrnl has an additional antifibrotic action and connected the skin fibrosis observed in skin biopsies of systemic sclerosis patients to lower concentrations of this molecule, which facilitated fibroblast activation [14]. Accordingly, loss of Metrnl led to worse cardiac fibrosis in knockout mice, whereas in vivo adenoviral overexpression of Metrnl had a protective effect on heart fibrosis [16].

### 2.3. Metrnl’s Role in Metabolic and Inflammatory Diseases

Meteorin-like protein (Metrnl) plays a dual regulatory role in metabolism by decreasing circulating fat and blood glucose levels. An in vitro study involving adipocyte-specific Metrnl-knockout mice demonstrated antagonistic activity against insulin resistance [16].

Different studies have investigated the association between Metrnl levels and insulin resistance. For instance, a case–control analysis involving patients with polycystic ovary syndrome (PCOS) revealed significantly reduced serum Metrnl levels, which were inversely associated with fasting insulin and glucose levels [30]. Similarly, many research teams have sought to elucidate the relationship between Metrnl and T2DM. While investigations by Zheng et al. [31] and Lee et al. [32] have established a negative association between the variables, the findings by Chung et al. [33] and Wang et al. [34] contradict this association, suggesting that Metrnl may exacerbate hyperinsulinemia and insulin resistance by inhibiting PPARγ. Additionally, Metrnl’s ability to attenuate endothelial dysfunction by PPARγ and AMPK signaling pathways emphasizes its therapeutic importance in metabolic disorders, where LPS-induced injury has been observed.

Metrnl improves myocardial ischemia–reperfusion damage (MI/RI) by polarizing M2 macrophages through AMPK [35]. In allergic airway models, Metrnl reduces inflammatory mediators by inhibiting the IKK/IκB/NF-κB pathway [36].

Metrnl has been consistently shown to exhibit an inverse association with inflammatory markers such as CRP and high-sensitivity CRP (hs-CRP) in a range of metabolic and cardiovascular conditions, including coronary artery disease (CAD), PCOS, gestational diabetes, and T2DM [16]. Wang et al. further reported a downregulated relationship of circulating Metrnl with the severity of diabetic nephropathy, particularly among patients with macroalbuminuria, indicating a potential link to impaired renal function [37]. Additionally, Metrnl levels were inversely associated with several renal and metabolic parameters, including blood urea nitrogen, serum creatinine, uric acid, albumin-to-creatinine ratio (ACR), disease duration, glycosylated hemoglobin (HbA1c), and use of renin–angiotensin–aldosterone system (RAAS) blockers [37]. Furthermore, Lin et al. suggested that increased Metrnl expression in the kidney may reduce renal injury by inhibiting the TGF-β1/Smads signaling pathway and decreasing α-SMA expression [38]. Metrnl expression appears to be upregulated in multiple inflammatory diseases, including RA, psoriasis, actinic keratosis, and prurigo nodularis [10]. Conversely, decreased expression of this protein has been observed in Graves’ disease [39]. Furthermore, Metrnl plays a pivotal role in cold-induced thermogenesis by promoting the browning of white adipose tissue, thereby enhancing energy expenditure and supporting thermal homeostasis [13].

Obesity is characterized by excessive accumulation of fatty acids and inflammatory mediators, particularly derived from visceral adipose tissue, which contributes to a persistent inflammatory state that promotes insulin resistance and the onset of T2DM. Additionally, evidence suggests that obesity may induce inflammatory responses within skeletal muscle, as indicated by infiltration of intermyocellular and perimuscular adipose tissue. Conversely, physical exercise exerts strong anti-inflammatory effects, potentially mediated through the upregulation and secretion of Metrnl during muscle contraction, which could be used to counteract the deleterious effects of obesity [40]. Metrnl has emerged as a candidate biomarker for metabolic disturbances, as its circulating levels have been associated with adverse lipid parameters. Specifically, positive correlations have been reported between Metrnl and high-density lipoprotein cholesterol (HDL-C).

In contrast, inverse relationships have been observed with low-density lipoprotein cholesterol (LDL-C), triglycerides, and total cholesterol [16]. Furthermore, a study involving Chinese individuals with T2DM demonstrated a significant association between serum Metrnl levels and visceral fat obesity (VFO) [41]. Lower serum Metrnl concentrations were noted in VFO groups compared with non-VFO groups, leading researchers to propose Metrnl as a noninvasive biomarker of VFO [41].

Metrnl has also been attributed to cytotropic effects, as it is involved in tissue repair by promoting cell division, differentiation, and maturation [13]. A recent study described it as a negative predictor of de novo heart failure in heart failure with preserved ejection fraction [42]. Another study demonstrated its inhibitory effect on cardiac remodeling with respect to hypertrophy and fibrosis [43,44]. Giden and Yasak highlighted the role of Metrnl in the diagnosis and treatment of acute ischemic stroke (AIS) [45]. They also reported an inverse correlation between circulating Metrnl levels and both serum troponin concentrations and the time elapsed from the onset of acute chest pain to hospital admission in individuals diagnosed with acute coronary syndrome (ACS) [45]. Moreover, elevated Metrnl concentrations have been associated with unfavorable clinical outcomes following ST-elevation myocardial infarction and may serve as a prognostic indicator of overall mortality during the acute phase [46].

A prior investigation explored the correlations among body mass index (BMI), Metrnl, and proinflammatory cytokines within the framework of inflammatory bowel disease (IBD) [47]. The study revealed significantly decreased Metrnl concentrations in patients with ulcerative colitis (UC) and Crohn’s disease (CD) compared to healthy controls. At the same time, levels of TNF-α and IL-6 were markedly elevated in the IBD cohort. Furthermore, an inverse correlation was identified between Metrnl concentrations and body mass index (BMI), collectively suggesting a potential pathogenic involvement of Metrnl in IBD pathophysiology [47]. At about the same time, Metrnl injection into IL-10^−/−^ mice was shown to have a beneficial effect in CD by exerting an anti-inflammatory effect on adipocytes and promoting their differentiation into mesenteric adipose tissue [48]. Beyond its immunometabolic functions, Metrnl has been proposed to contribute to neuroprotective mechanisms due to its structural homology with meteorin—a protein known to regulate neuronal differentiation and axonal growth—and its capacity to traverse the BBB [14]. In line with this, Berghoff et al. presented evidence supporting Metrnl’s role within the muscle–brain axis, which could possibly be enhanced by exercise and muscle contraction and directly affect CNS function and/or plasticity [15]. Another randomized controlled trial linking exercise to reduced inflammation and improved physical and cognitive outcomes in older adults also supports these findings [49].

The statistically significant correlation between serum and cerebrospinal Metrnl concentrations and the fact that the albumin/CSF/serum ratio significantly correlates with the Metrnl/CSF/serum ratio indicate a high degree of both basal and inducible CMB permeability for this adipomyokine, compared to classical adipokines [15].

The accumulated data demonstrate that Metrnl functions as a universal immunomodulator, utilizing AMPK-PPARγ and AMPK-PAK2 signaling pathways. Overall, this leads to decreased activation of NF-κB, diminished NLRP3 inflammasome assembly, and reduced oxidative stress levels, resulting in protection against apoptosis and fibrosis in cardiovascular and pulmonary tissues. At the same time, its expression in barrier and stromal tissues, as well as its modulation in autoimmune diseases such as psoriasis, RA and IBD, highlights its role in innate defense and tissue repair. When Metrnl levels are elevated in the blood, markers of inflammation and other biomarkers (e.g., CRP, blood sugar, or markers of kidney damage) are lower, indicating an inverse correlation [49].

The measurement of Metrnl levels in blood provides potential value for disease monitoring of heart conditions, kidney and metabolic disorders. The molecule serves as a “bridge” that connects inflammation, metabolic processes and tissue remodeling. The variable disease-specific nature of Metrnl effects requires extensive studies before it becomes ready for clinical use [49]

### 2.4. Potential Diagnostic and Therapeutic Implications

As already illustrated, Metrnl may have a dual role, both in the initial diagnosis and in the treatment of various pathologies. Concerning its diagnostic significance, Metrnl has the potential to act as a predictive biomarker for various cardiac and metabolic conditions. In certain instances, its levels may correlate with disease activity. This is true for T2DM, polycystic ovary syndrome (PCOS), insulin resistance, visceral obesity, coronary artery disease (CAD), intensive angiography (IIA), and heart failure, as well as inflammatory diseases, including allergic asthma, psoriatic arthritis, osteoarthritis, Crohn’s disease, ulcerative colitis, and Graves’ disease [11,16,30,31,32,33,34,37,39,41,42,43,44,45,46,47].

Similarly, Metrnl may prove useful in a number of other diseases. In particular, its anti-inflammatory and putative antifibrotic effects could be used in various ways to combat immune, inflammatory, and metabolic disorders. For example, promising data suggest its use in the treatment of myocardial infarction [29], osteoporosis, and traumatic fractures [12,13,16]. Metrnl also appears to have a role in the treatment of T1DM. Yao et al. have shown promising results after intravenous administration of Metrnl in 4-week-old non-obese and diabetic (NOD) mice [50]. Intravenous infusion of Metrnl may postpone the development of diabetes and alleviate lymphocyte infiltration into pancreatic islets, thus opening new avenues for the treatment of patients with T1DM. Although Huang et al. characterized Metrnl as an essential protein for skeletal development, based on the lack of observed in vivo changes in skeletal parameters after loss of Metrnl expression. The osteoinductive effects of Metrnl in vitro represent a promising mechanism for manipulating osteoblast differentiation in cell-based orthopedic therapies for fracture healing [12].

Another interesting effect of Metrnl is its action in improving age-related cognitive impairment, as well as age-related muscle damage and inflammatory myopathies [16]. Lee et al. effectively showed that administering Metrnl via intramuscular injection can stimulate muscle regeneration in aged tissue by promoting TNF-α–dependent apoptosis of fibro/adipogenic progenitor cells [51]. Similarly, Metrnl has been suggested to play a role in improving age-related cognitive dysfunction, based on a study conducted in D-galactose-induced aging models [16,52]. Regarding the promising role of Metrnl in the treatment of heart disease, it has been described that Metrnl infusion can promote angiogenesis after myocardial infarction, as observed in mouse models [16]. Topical administration of Metrnl has been proposed as another approach to stimulate angiogenesis and wound epithelialization in diabetic patients with poor wound healing and ulcer complications [45,53].

The effects of Metrnl in neoplastic diseases appear to depend on the microenvironmental conditions. Research indicates that Metrnl exhibits dual functions in cancer, as it may act as both an anti-inflammatory agent and a tumor growth regulator. However, current evidence suggests that Metrnl secretion in the tumor microenvironment may promote tumor progression [24,54]. The dual nature of Metrnl in oncology necessitates that researchers study its effects in relation to the tumor microenvironment and context-dependent conditions. A study conducted in mice with allergic asthma showed that Metrnl inhibits the development and activity of dendritic cells, hence impairing type 2 inflammatory responses [55].

Finally, the neuroprotective role of Metrnl also deserves further investigation, as it has the potential to influence neuronal plasticity and directly influence growth processes and migration in the CNS [14]. Metrnl is a multifunctional mediator with different roles as an adipokine, cytokine, myokine, and neurotrophic factor (Figure 1). This enables Metrnl to serve as a prognostic indicator and a novel immune biomarker.

## 3. Follistatin-like 1 (FSTL1) and Its Role in Inflammation in Autoimmune Rheumatic Diseases

### 3.1. Structure and Function of FSTL1

Follistatin-like 1 (FSTL1) is a small glycoprotein primarily synthesized by mesenchymal-derived cells and belongs to the SPARC (secreted protein acidic and rich in cysteine) family of matricellular proteins [56,57,58]. A comparative analysis of the mouse (GenBank Q62356; 306 amino acids) and human (GenBank Q12841; 308 amino acids) protein sequences reveals a high degree of similarity, with 272 amino acids conserved (94.4%). However, the secretion signal shows considerable variability, and glycosylation patterns differ depending on the cell type [56,59].

FSTL1 is involved in various processes, including cell growth, tissue remodeling, repair, immune modulation, cell proliferation, differentiation, apoptosis, and migration [60,61,62,63]. It contributes to the development of the CNS, lungs, ureter, and skeletal system and even in cardiac myocytes [64,65,66,67] and is implicated in the regulation of tumor growth and metastasis [68,69].

### 3.2. FSTL1 in Autoimmune and Chronic Inflammatory Diseases

FSTL1 has been implicated in the pathology of RA [70]. Tanaka et al. first identified it when FSTL1 autoantibodies were detected in the synovial fluid and serum of patients with RA [71]. In mouse studies, mRNA of *FSTL1* was also demonstrated to be associated with increased activity of several proinflammatory cytokines [72,73,74]. There is evidence of a correlation between serum levels of FSTL1 and the progression of autoimmune rheumatic diseases [75,76].

In recent years, the investigation into the role of FSTL1 in the etiology of osteoarthritis has been increasingly studied. Elevated serum concentrations of FSTL1 have been noted in osteoarthritis and Sjögren’s syndrome [75,77]. Its overexpression promotes the production of inflammatory mediators, which impair chondrocyte differentiation and contribute to the progression of osteoarthritis. In the serum of RA patients, antibodies targeting FSTL1 are more commonly detected than in those with Sjögren’s disease [75], whereas such antibodies have not been identified in individuals with osteoarthritis [71].

Kawasaki disease is believed to be triggered by an infection that causes inflammation of the blood vessels. Plasma levels of FSTL1 are significantly elevated in patients with this disease, but they gradually decrease with immunoglobulin therapy [78]. Increased expression of FSTL1 has been documented in individuals diagnosed with asthma, pulmonary fibrosis, and chronic obstructive pulmonary disease [79]. In mouse models, Fstl1 mRNA expression has been identified in lung mesenchymal cells, respiratory smooth muscle cells, goblet epithelial and endothelial cells [57]. If its function is impaired, neonates die from respiratory distress [80].

Diseases such as liver cirrhosis, idiopathic pulmonary fibrosis, endomyocardial fibrosis, systemic sclerosis and other diseases are associated with organ fibrosis [78,79,80,81,82,83]. It results from the induction of cytokines such as TGF-β1, which damage the epithelium and lead to increased collagen production [84].

FSTL1 binds to disco-interacting protein 2 homolog A (DIP2A) on endothelial cells, driving Smad2/3-mediated angiogenesis independently of canonical TGF-β receptors [85]. It participates in liver fibrogenesis by modulating fibrosis through the DIP2A-Smad/JNK axes and the TGF-β1–miR-29a circuit [86]. Data indicate that the FSTL1/TGF-β1 signaling cascade underlies profibrotic responses in the pathology of these and other autoimmune diseases, making FSTL1 a possible target for therapeutic intervention [63,70]. In hypoxia-challenged pulmonary smooth muscle cells, FSTL1 inhibits ERK phosphorylation and modulates AMPK, attenuating proliferation and migration [87].

Overall, the evidence confirms that FSTL1 is a matricellular glycoprotein that performs multiple roles. Its structure remains conserved, but in different cell types, its glycosylation patterns differ and affect its expression levels and activity. The presence of FSTL1 in RA patients makes it a promising biomarker and highlights its potential as a factor contributing to the pathogenesis of the disease. Its elevated levels of FSTL1 in Kawasaki disease, asthma, pulmonary fibrosis and systemic sclerosis indicate its involvement in vascular and fibrotic remodeling processes through the TGF-β1–FSTL1 signaling pathway. All research results demonstrate that FSTL1 functions as a key mediator of inflammation and fibrosis, which makes it a promising therapeutic target for systemic autoimmune and chronic inflammatory diseases.

### 3.3. Therapeutic Potential of Targeting FSTL1

The therapeutic potential of FSTL1 is linked to various diseases. Recombinant FSTL1 administration may help reverse myocardial death following myocardial infarction in humans [88,89]. FSTL1 inhibition could attenuate fibroblast activation and reduce lung fibrosis [90]. FSTL1 regulates chondrocyte proliferation and chondrogenic differentiation of mesenchymal stem cells during inflammation [91].

In RA, FSTL1 enhances the production of proinflammatory cytokines, such as TNF-α, IL-6, and IL-1β, thereby amplifying inflammatory cascades. Targeting FSTL1 could reduce synovial inflammation and joint destruction in RA. Epitope-specific monoclonal neutralizing antibodies (nAbs) for blocking FSTL1 are one of the potential treatments for multiple organ fibrosis and systemic autoimmune diseases [63]. This approach will lead to the blockage of FSTL1 activity at the protein level, thereby suppressing inflammation and joint damage.

The use of siRNA or antisense oligonucleotides will lead to the silencing of FSTL1 gene expression, resulting in long-term control of disease progression in arthritis [92].

Currently, there are no effective treatments for systemic autoimmune diseases and organ fibrosis. Targeting FSTL1 may present new therapeutic opportunities and improve patient outcomes. Nonetheless, further research is needed to validate FSTL1′s role in these conditions and to fully assess its therapeutic potential.

The figure below illustrates the multifaceted biological roles of Follistatin-Like 1 (FSTL1) (Figure 2).

In conclusion, scientific data show that Metrnl and FSTL1 are secreted proteins that connect immune responses, metabolic processes, and tissue remodeling activities through different mechanisms. Metrnl acts as an immunomodulator, depending on the situation, while FSTL1 functions as a matrix–cell cytokine. Both molecules activated some identical signaling pathways, such as AMPK regulation, angiogenic, and inflammatory cascades. Their complementary yet opposing functions suggest that Metrnl may primarily serve as a potential biomarker for metabolic and vascular disorders. In contrast, FSTL could be a therapeutic target for autoimmune rheumatic and fibrotic diseases.

Table 1 presents the primary sources of production, key signaling pathways, immunological effects, tissue outcomes, clinical associations, and therapeutic implications of Metrnl and FSTL1.

In terms of their therapeutic potential, it can be said that both molecules hold promise for treating various diseases, but their roles vary depending on the situation. Metrnl can serve as both a biomarker and a therapeutic enhancer for tissue repair. FSTL1 is specifically associated with a certain condition, and its inhibition may reduce inflammation and fibrosis. Its evaluation regarding the regenerative role requires additional research.

## 4. Metrnl and FSTL1 in Other Autoimmune Pathologies: Role in Gastrointestinal Inflammatory Diseases

In addition to their established roles in metabolic, cardiovascular and musculoskeletal systems, Metrnl and FSTL1 are also involved in other autoimmune pathologies that affect the gastrointestinal tract [93]. The scientific data support the potential use of these organokines in monitoring diseases and developing specific therapeutic approaches.

### 4.1. The Role of Metrnl in Colorectal Cancer (CRC)

Uzun et al. examined the immunoreactivity of Metrnl, asprosin, and irisin at different stages of differentiation in 60 cases of colorectal adenocarcinomas [94]. They were diagnosed between 2000 and 2020, with 20 of them equally presenting well-differentiated, moderately differentiated, and poorly differentiated cancers at each stage of differentiation. Metrnl immunoreactivity exhibited no significant difference between normal colonic mucosa and well-differentiated adenocarcinomas [94]. Conversely, Uzun et al. reported a statistically significant rise in Metrnl immunoreactivity in moderately differentiated adenocarcinomas compared to normal colonic mucosa (*p* < 0.048), and a significant decrease in poorly differentiated cancers compared with moderately differentiated adenocarcinomas [94]. In another study, Onat et al. observed significantly lower Metrnl immunoreactivity in colon carcinoma tissues than in healthy tissues [95].

### 4.2. The Role of Metrnl in Inflammatory Bowel Diseases (IBDs)

In a case–control study, Gholamrezayi et al. investigated the association between serum Metrnl and IBD [47]. The study cohort included 85 IBD patients—comprising 42 individuals with ulcerative colitis (UC) and 43 with Crohn’s disease (CD)—alongside 54 healthy controls. The findings revealed that serum Metrnl concentrations were significantly reduced in IBD patients compared to the control group [47]. Furthermore, an inverse association was noted between circulating Metrnl levels and proinflammatory cytokines IL-6 and TNF-α, as well as BMI in both UC and CD patients [47].

A protective role for Metrnl has been shown in mouse models of dextran sodium sulfate (DSS)-induced colitis [96]. After administration of 3% DSS water, Metrnl knockout mice exhibited more severe colitis compared with wild-type (WT) control mice. Zhang et al. observed significantly higher serum levels and mRNA expression of IL-6, IL-1β, and TNF-α in the colon of Metrnl^−/−^ mice compared with WT mice after 3% DSS treatment [97]. The authors concluded that Metrnl appears to be a promising therapeutic target for ulcerative colitis [97].

Metrnl may have useful clinical applications for patients with Crohn’s disease [96]. Zuo et al. examined Metrnl expression in mesenteric adipose tissue (MAT) obtained from 21 patients without Crohn’s disease (CD) who underwent surgery for colon cancer, along with 16 patients with CD who had an initial ileocecal resection due to stenosis [48]. The authors reported significantly higher Metrnl expression in MAT of CD patients compared to controls [48]. A compensatory anti-inflammatory response could explain this high Metrnl expression in MAT of CD patients. In line with the human study, Zuo et al. found higher Metrnl expression in MAT of Il-10^−/−^ mice compared to MAT of WT mice [48]. By activating the STAT5/PPAR-γ signaling pathway, Metrnl treatment promotes adipocyte differentiation in MAT, alleviates inflammation in MAT, reduces mesenteric hypertrophy, and improves adipocyte intrinsic function [48].

### 4.3. The Role of FSTL1 in CRC

The role of FSTL1 in the pathogenesis of CRC is complex and not yet fully understood [98]. Zhao et al. investigated the expression levels of FSTL1 in peripheral plasma and tissues of colorectal cancer patients [99]. They reported significantly higher serum levels of FSTL1 in patients with colorectal cancer compared to healthy controls. Patients exhibiting high FSTL1 levels in fully scanned tumor tissue had significantly reduced overall survival at 3 years compared to those with low FSTL1 expression. Notably, lower FSTL1 levels within the tumor stroma were associated with worse long-term survival outcomes [99].

In a Chinese study, Gu et al. demonstrated significantly higher expression of FSTL1 in colorectal cancer tissues than in adjacent non-tumor tissues [100]. Elevated FSTL1 levels in cancer tissue were linked to lymph node metastasis, poor prognosis, and increased tumor invasion depth. Studies have shown that FSTL1 contributes to the promotion of cell invasion, migration, and metastasis in colorectal cancer (CRC) [100].

Bevivino et al. reported that FSTL1 is upregulated in both sporadic human CRC samples and CRC cell lines [101]. In these cells, silencing FSTL1 resulted in decreased proliferation and increased sensitivity to chemotherapeutic agents, such as oxaliplatin and irinotecan [101].

Research findings demonstrate that FSTL1 exhibits different effects due to compartmental heterogeneity and the chosen analytical technique. The complex role of FSTL1 in colorectal cancer requires further studies to elucidate its complete involvement and mechanism. The prognostic assessment of FSTL1 remains unclear because its effects depend on its specific localization.

### 4.4. The Role of FSTL1 in Gastric Cancer

Li et al. found that FSTL1 is significantly overexpressed in gastric cancer tissue compared to normal mucosa and that elevated FSTL1 levels are linked to poorer prognosis in gastric cancer patients [102]. The expression of FSTL1 showed a strong correlation with tumor–node–metastasis (TNM) stage, tumor size, and lymph node involvement [102]. Sierzega et al. investigated the role of circulating myokines in the development of cancer-associated cachexia (CAC) among gastric cancer patients [103]. They observed notably higher FSTL1 levels in the peripheral blood of patients with CAC in comparison to controls. However, FSTL1 in peripheral blood was found to be a poor predictor of cachexia and is not associated with tumor stage [103].

Peng et al. demonstrated that FSTL1 expression is significantly elevated in gastric cancer cell lines compared to normal gastric mucosal epithelial cells [104]. Suppressing FSTL1 in these gastric cells led to cytotoxic effects, as evidenced by increased lactate dehydrogenase release and reduced cell viability. Furthermore, their findings showed that FSTL1 inhibition enhances apoptosis in gastric cell lines, evidenced by a substantial increase in caspase-9 and caspase-3 activity [104].

### 4.5. The Role of FSTL1 in Liver Diseases

FSTL1 plays a role in the progression of liver neoplasms and fibrosis [105]. In mouse models of chronic liver disease, hepatic FSTL1 mRNA levels are also elevated [106]. Li et al. observed that FSTL1 expression in liver tissue rose progressively with advancing stages of liver fibrosis, and plasma FSTL1 concentrations were utilized to assess the diagnostic accuracy for detecting liver fibrosis [107]. According to these findings, plasma FSTL-1 levels may serve as a noninvasive biomarker for advanced liver fibrosis, reducing the need for liver biopsies [107].

### 4.6. The Role of FSTL1 in IBD

The role of FSTL1 in intestinal inflammation is still not fully understood [97]. In a Chinese study, Li et al. demonstrated that FSTL1 could aggravate dextran sulfate sodium (DSS)-induced colitis [108]. The authors reported that overexpression of FSTL1 promoted proinflammatory M1 polarization of macrophages. They revealed that serum FSTL1 levels were significantly elevated in individuals with active ulcerative colitis compared to both healthy controls and individuals with inactive disease [108]. In another study, Li et al. examined serum levels of FSTL1 in patients with systemic autoimmune diseases, including 20 patients with Crohn’s disease and 22 with UC [75]. Compared with healthy controls, the serum concentration of FSTL1 was significantly higher in patients with ulcerative colitis than in those with Crohn’s disease. A small difference in serum FSTL1 concentration was found between patients with Crohn’s disease and healthy subjects [75]. Bai et al. found significantly increased serum FSTL1 levels in individuals with active UC compared to healthy subjects [109]. Therefore, FSTL1 may also represent a promising therapeutic target for UC [108].

In Table 2, we summarize key human studies. A full quantitative meta-analysis is beyond the scope of our research; however, we highlight where effects are available and consistent, even in the face of heterogeneous data.

The table integrates clinical and translational findings, disease context, direction of effects, and statistical signal, preventing misinterpretation as the data are diverse and sometimes contradictory. Overall, this structured summary enhances the integration of evidence and highlights the translational relevance of these molecules.

## 5. Translational Considerations and Safety, Associated with Metrnl and FSTL1

Preclinical testing of therapeutic candidates is primarily conducted in animal models, as demographic and other characteristics vary in humans and pose limitations. Human cell in vitro models help to solve some of them, but they do not replicate the complete biological reactions that occur in vivo. Translational concerns with both molecules are primarily related to pleiotropic effects, gaps in knowledge about some of their receptors, half-lives, and doses, as well as potential oncological risks associated with their dual role.

Both molecules are expressed in different tissues and are involved in various signaling pathways, which can lead to unwanted systemic effects. Metrnl generally has positive anti-inflammatory and metabolic effects. Still, there is evidence that its secretion into the tumor microenvironment can suppress the activity of CD8^+^ T cells by disrupting mitochondrial function and facilitating tumor progression [54].

FSTL1 exerts a protective role in tissue repair. Still, there is evidence that it can stimulate fibroblast activation and profibrotic remodeling through DIP2A-Smad/JNK signaling. The JNK/ERK pathway is also involved, and as a result, excessive deposition of extracellular matrix mass, tissue stiffness, and fibrosis can occur [86]. The therapeutic approach of targeted delivery or context-specific agonism (activation)/antagonism (blockade) needs to be applied to specific tissues, cell types, and diseases. The aim is to achieve optimal therapeutic effects with minimal pleiotropic side effects. For Metrnl, systemic activation may be beneficial in metabolic or vascular diseases, but not in cancer conditions. The inhibition of FSTL1 shows promise for treating fibrotic and autoimmune diseases; however, blocking it throughout the body may disrupt its roles in angiogenesis and development [86].

The knowledge gaps about the receptors that both molecules use need to be filled. The goal is to gain a comprehensive understanding of how and in which signaling pathways Metrnl and FSTL1 specifically intervene, as this will enable the treatment to be precisely targeted and minimize unwanted effects in off-target tissues and cells. For Metrnl, for example, the non-endothelial receptors and coreceptors (e.g., in muscle, fat or immune cells) are still undefined.

For FSTL1, data indicate that its primary partner is DIP2A, which activates Smad2/3 and JNK/ERK signaling in various tissues. However, it can also interact with alternative partners (e.g., TLR4, BMP receptors) in different tissues and enhance inflammation [58,110]. This means that the mechanism of action of FSTL1 may change depending on the tissue context. Due to this variability, not all patients will respond equally to therapy targeting FSTL1. It is necessary to consider receptor expression profiling in clinical applications to avoid negative outcomes.

Scientists are developing different methods to produce cytokine treatments. The aim is safer, longer-lasting, and more targeted therapies. This is because many molecules have short half-lives, can sometimes cause severe systemic inflammation at high doses, and often operate within a very narrow therapeutic window [111,112]. The process of quantification is critical because any deviation from the right amount will lead to a shift from beneficial to harmful effects. To overcome these problems, researchers are developing and testing various possible options (e.g., immunocytokines, sustained-release formats, receptor-dependent muteins, nanoparticles, vectors or other forms to deliver to the target organ/cells). For Metrnl and FSTL1, which act in multiple tissues and pathways, all of these approaches could be applied with the aim of minimizing side effects and maximizing therapeutic benefits [112].

The dual role of FSTL1 and Metrnl, as mentioned earlier, should be considered in therapeutic strategies. Especially for FSTL1, as it may facilitate tumor progression in cancer conditions. The following important points need attention when interpreting clinical data. The lack of standardization in reagents, calibration methods and sample handling procedures makes it challenging to perform direct comparisons between different studies. A significant limitation in the development of Metrnl as a biomarker or therapeutic candidate is the lack of standardized assays for circulating measurements, as different ELISA kits have yielded varying concentration ranges and conflicting associations with metabolic and inflammatory diseases [31,113].

Literature results indicate that Metrnl levels are also influenced by different factors (renal function, exercise, and metabolic data), which complicates the interpretation of pharmacokinetic data [15,114]. The achievement of early-phase clinical trials requires researchers to establish specific assay platforms, use standardized biological sample collection methods and protocols, and control over other variables to achieve cohort comparability. This will enhance the robustness of research and facilitate reliable and precise assessment of Metrnl and FSTL1 as potential therapeutic targets.

## 6. Conclusions

Metrnl and FSTL1 have emerged as valuable biomarkers owing to their elevated expression in different inflammatory, autoimmune and metabolic diseases. They have been linked with CRC, IBD, gastric cancer and liver fibrosis, as well as conditions such as RA, psoriasis, acute coronary syndromes, myocardial infarction, pulmonary fibrosis, asthma and Kawasaki disease, etc. These data suggest their potential use for monitoring inflammatory activity, prognostic utility, and even survival outcomes. They may reflect the possible severity of the disease, to clarify disease progression and guide a suitable medical approach/therapy. Their measurable presence in blood, serum, and tissue samples makes Metrnl and FSTL1 suitable for noninvasive diagnostic or prognostic testing. This makes them promising biomarkers not only for identifying inflammatory and autoimmune diseases, but also for guiding therapeutic interventions with additional potential.

Some issues need to be addressed, as some studies have inconsistent results in clinical testing. The mechanisms by which Metrnl and FSTL1 exert their effects, as well as the pathways and their receptors, need to be fully elucidated. Moreover, they are widely distributed in various organs and tissues throughout the body, and multiple factors influence their levels. Further well-controlled and comprehensive studies are needed to evaluate their use as predictive biomarkers and as potential therapeutic targets.

## Figures and Tables

**Figure 1 ijms-26-09711-f001:**
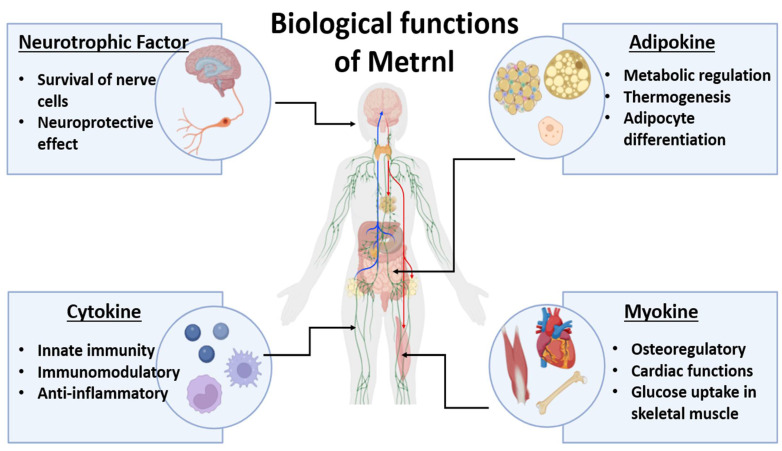
The biological roles of Meteorin-like protein (Metrnl). Metrnl is a multifunctional secreted protein that exerts diverse biological effects through its roles as a neurotrophic factor, adipokine, cytokine, and myokine. As a neurotrophic factor, Metrnl supports the survival of nerve cells and has neuroprotective effects. Acting as an adipokine, it regulates metabolism, thermogenesis, and adipocyte differentiation, contributing to energy homeostasis. Through its cytokine functions, Metrnl participates in innate immunity, exhibits immunomodulatory activity, and exerts anti-inflammatory effects. Finally, as a myokine, Metrnl is involved in bone regulation, cardiac function, and glucose uptake in skeletal muscle, linking it to both musculoskeletal and cardiovascular health. Collectively, these pleiotropic functions underscore the emerging role of Metrnl in metabolic, inflammatory, and autoimmune diseases.

**Figure 2 ijms-26-09711-f002:**
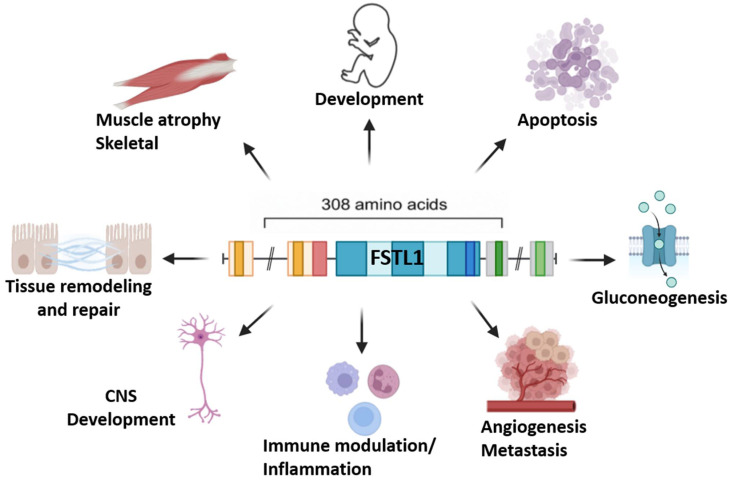
Key biological functions of Follistatin-Like 1 (FSTL1). FSTL1 is a multifunctional glycoprotein that is involved in diverse physiological and pathological processes. It contributes to embryonic development and central nervous system (CNS) maturation, while also playing a role in skeletal muscle atrophy and regeneration. FSTL1 regulates tissue remodeling and repair, influences apoptosis, and modulates gluconeogenesis through metabolic pathways. Furthermore, it is a key mediator of immune modulation and inflammation, and it has been implicated in angiogenesis and metastasis across different cancer models. The pleiotropic actions of FSTL1 highlight its importance as both a regulator of homeostatic functions and a potential therapeutic target in diverse disease contexts.

**Table 1 ijms-26-09711-t001:** Distinct and overlapping roles of Meteorin-like (Metrnl/IL-41) and Follistatin-like 1 (FSTL1) in immunity, metabolism, and fibrosis.

Features	Metrnl (Meteorin-Like, IL-41)	Refs.	FSTL1 (Follistatin-Like 1)	Refs.
Primary site of expression/production	Barrier tissues (skin, mucosa) and stromal cells (fibroblasts, endothelial cells, adipocytes, etc.)	Li et al., 2023 [16] Uschach et al., 2015 [11]	Mainly by mesenchymal-derived cells, including fibroblasts, chondrocytes, smooth muscle cells, etc.	Mattiotti et al., 2018 [58]; Karisa et al., 2025 [85]
Major signaling pathways	AMPK–PPARγ, AMPK–PAK2, inhibition of NF-κB, STAT5/PPARγ	Li et al., 2023 [16]; Chen et al., 2025 [35]	DIP2A–Smad2/3, TGF-β1–Smad/JNK/ERK, AKT/ERK	Karisa et al., 2025 [85]; Xu et al., 2020 [29]
Immune effects	M2 macrophage polarization reduces ROS and NLRP3 inflammasome activity	Chen et al., 2025 [35]; Khodir et al., 2025 [36]	Enhances TNF-α, IL-6, IL-1β; amplifies autoimmune inflammation response	Tanaka et al., 1998 [71]; Li et al., 2011 [75]
Tissue outcomes	Protects against fibrosis, myocardial ischemia–reperfusion injury, and endothelial inflammation	Chen et al., 2025 [35]	Promotes fibrosis in the lung, liver, and systemic sclerosis; drives cartilage degradation in RA/OA	Xu et al., 2020 [29]; Fujimoto et al., 2016 [81]
Clinical associations	Linked with CAD, T2DM, PCOS, diabetic nephropathy, IBD, ischemic stroke	Wang et al., 2019 [34]; Giden and Yasak 2023 [45]	Elevated in RA, OA, Sjögren’s, systemic sclerosis, pulmonary fibrosis, and Kawasaki disease	Karisa et al., 2025 [85]
Therapeutic implications	Candidate biomarker and protective cytokine; potential in regenerative and metabolic disorders	Li et al., 2023 [15]; Jamrasi et al., 2025 [49]	Target for fibrosis and autoimmunity; approaches include neutralizing antibodies, siRNA, and antisense therapies	Xu et al., 2020 [29]; Mattiotti et al., 2018 [58]

(RA—Rheumatoid arthritis; OA—Osteoarthritis; CAD—Coronary artery disease; T2DM—Type 2 diabetes mellitus; PCOS—Polycystic ovary syndrome; IBD—Inflammatory bowel disease).

**Table 2 ijms-26-09711-t002:** Summary of key human studies on Metrnl and FSTL1 in inflammatory, metabolic, cardiovascular, and autoimmune diseases.

Molecule	Disease/Context	Population	Direction vs. Controls/Outcome	Significance	References
Metrnl	Autoimmune skin/joint tissues	RA synovial membrane & psoriatic skin biopsies	↑ Metrnl tissue expression vs. non-inflamed controls	*p* < 0.05	Uschach et al., 2015 [11]; Li et al., 2023 [16]
Metrnl	PCOS	Case–control women	↓ serum Metrnl; inverse with insulin/glucose	*p* < 0.05	Zheng et al., 2018 [31]; Li et al., 2023 [16]
Metrnl	Type 2 diabetes mellitus (T2DM)	Mixed cohorts	Conflicting: ↓ Metrnl in some cohorts; ↑ Metrnl in others	Mixed	Zheng et al., 2018 [31]; Wang et al., 2019 [34]
Metrnl	Diabetic nephropathy (T2DM)	Adults with T2DM	↓ Metrnl with ↑ nephropathy severity; inverse with eGFR	*p* < 0.01	Wang et al., 2019 [34]; Li et al., 2023 [16]
Metrnl	Coronary artery disease (CAD)	Adults with CAD	Lower levels are associated with adverse risk profiles	*p* < 0.05	Wang et al., 2019 [34]; Li et al., 2023 [16]
Metrnl	Acute coronary syndrome (ACS)	Adults with ACS (2021–2023)	Inverse correlation between circulating Metrnl levels and serum troponin concentrations;	*p* < 0.05	Giden and Yasak, 2023 [45]; Li et al., 2023 [16]
Metrnl	STEMI (prognosis)	STEMI patients (Cohorts 2021–2024)	Higher admission Metrnl linked to worse outcomes/mortality	*p* < 0.05; independent in models	Li et al., 2023 [16]
Metrnl	Ischemic stroke	Adults with AIS	↓ Metrnl vs. controls; diagnostic/prognostic signal	*p* < 0.01	Giden and Yasak, 2023 [45]; Li et al., 2023 [16]
Metrnl	IBD (UC/CD)	IBD cohorts (2018–2022)	↓ Metrnl vs. controls; inverse with TNF-α/IL-6; inverse with BMI	*p* < 0.01	Li et al., 2023 [16]
Metrnl	Neuro–metabolic axis	Patients/CSF cohorts	Serum– and CSF correlation; BBB permeability signal	*p* < 0.05	Berghoff et al., 2021 [15]; Li et al., 2023 [16]
FSTL1	Rheumatoid arthritis (RA)	RA patients	↑ FSTL1 autoantibodies (serum & synovial fluid)	*p* < 0.05	Tanaka et al., 2019 [83]; Li et al., 2011 [75]
FSTL1	Osteoarthritis (OA) & Sjögren’s	Adult cohorts	↑ serum FSTL1 in OA & Sjögren’s; RA > Sjögren’s for auto-Abs; absent in OA	*p* < 0.05	Fujimoto et al., 2016 [81]; Li et al., 2011 [75]
FSTL1	Kawasaki disease (pediatric vasculitis)	Pediatric KD	↑ plasma FSTL1 in acute phase; ↓ post-IVIG	*p* < 0.05	Fujimoto et al., 2016 [81]
FSTL1	Systemic sclerosis (SSc)	Adults with SSc	↑ circulating FSTL1; correlates with fibrosis severity	*p* < 0.01	Fujimoto et al., 2016 [81]; Xu et al., 2020 [29]
FSTL1	Pulmonary fibrosis/ILD	ILD cohorts	↑ serum/tissue FSTL1; association with progression	*p* < 0.05	Xu et al., 2020 [29]
FSTL1	Asthma/COPD (inflammatory lung disease)	Respiratory cohorts (2017–2021)	↑ circulating FSTL1 vs. controls	*p* < 0.05	Fujimoto et al., 2016 [81]

Legend: ↑—increased/upregulated; ↓—decreased/downregulated. Abbreviations: RA—rheumatoid arthritis; OA—osteoarthritis; PCOS—polycystic ovary syndrome; T2DM—type 2 diabetes mellitus; CAD—coronary artery disease; ACS—acute coronary syndrome; STEMI—ST-elevation myocardial infarction; AIS—acute ischemic stroke; IBD—inflammatory bowel disease; UC—ulcerative colitis; CD—Crohn’s disease; SSc—systemic sclerosis; ILD—interstitial lung disease; IVIG—intravenous immunoglobulin.

## Data Availability

Not applicable..

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
