# Peer review of "Emerging Therapeutic and Inflammation Biomarkers: The Role of Meteorin-Like (Metrnl) and Follistatin-Like 1 (FSTL1) in Inflammatory Diseases"

_ijms, 2025, doi:10.3390/ijms26199711_

Round 1

Reviewer 1 Report

Comments and Suggestions for Authors

The manuscript provides a thorough and well-referenced overview of Metrnl and FSTL1 as potential biomarkers and therapeutic targets in inflammatory, autoimmune, and metabolic diseases. The literature coverage is broad, and the structure is logical, moving from biological function to disease association and potential clinical applications. 

However, several areas would benefit from more synthesis, greater discussion of translational implications, and refinement of figure legends for improved clarity.

Major:

  1. Mechanistic insights not fully developed
    • There are many disease associations described in the manuscript, but they lack some extent of mechanistic explanations rather than briefly mentioning signaling pathways (e.g., AMPK, PPARδ, TGF-β etc).
    • Recommendation: Expand mechanistic discussion, possibly using schematic diagrams to show the pathway integration for both Metrnl and FSTL1.
  2. Therapeutic implications potentials and limitations
    • The manuscript presents therapeutic potentials, but limitations, and safety concerns are not discussed (e.g., challenges of systemic cytokine modulation, off-target effects).
    • Authors can consider adding a section to discuss about translational hurdles and required preclinical/clinical validation.
  3. Conflicting data explanation
    • While the manuscript cites multiple studies on Metrnl and FSTL1. There are some contradictory findings (e.g., Metrnl in type 2 diabetes: refs [25–28]). The Authors can describe the possible causes for discrepancies (population differences, assay types, disease stage, etc.).
  4. Studies analysis or summary
    • Given the large number of cited studies, a meta-summary (e.g., direction of effect, sample size, statistical significance) would strengthen the review readability.
    • I would recommend authors add a table summarizing human studies with disease type and main finding to help readers to follow through.

Minor:

  1. Typographical/Formatting issues

Some reference numbers are split across lines (e.g., [43]).

      2. Figure quality and caption detail

Figures 1 and 2 are schematic but lack sufficient mechanistic detail or citation of supporting references. Suggest revising figures to be more informative, showing pathways, receptors, and effects.

Author Response

Reviewer 1

Comments and Suggestions for Authors

The manuscript provides a thorough and well-referenced overview of Metrnl and FSTL1 as potential biomarkers and therapeutic targets in inflammatory, autoimmune, and metabolic diseases. The literature coverage is broad, and the structure is logical, moving from biological function to disease association and potential clinical applications.

Authors’ reply: We thank the Reviewer for the constructive comments and careful evaluation of our work. We carefully revised the manuscript and addressed all the concerns as outlined below.

However, several areas would benefit from more synthesis, greater discussion of translational implications, and refinement of figure legends for improved clarity.

Major:

Mechanistic insights not fully developed

There are many disease associations described in the manuscript, but they lack some extent of mechanistic explanations rather than briefly mentioning signaling pathways (e.g., AMPK, PPARδ, TGF-β etc).

Recommendation: Expand mechanistic discussion, possibly using schematic diagrams to show the pathway integration for both Metrnl and FSTL1.

Authors’ reply: We agree with the Reviewer’s observation. We expanded the discussion of mechanistic pathways for both Metrnl and FSTL1, including their interactions with AMPK, PPARδ, TGF-β, and other key signaling cascades.

Therapeutic implications potentials and limitations

The manuscript presents therapeutic potentials, but limitations, and safety concerns are not discussed (e.g., challenges of systemic cytokine modulation, off-target effects).

Authors can consider adding a section to discuss about translational hurdles and required preclinical/clinical validation.

Authors’ reply: We added a new subsection highlighting translational hurdles, including safety concerns related to systemic cytokine modulation, potential off-target effects, and the necessity for preclinical and clinical validation. This addition balanced the discussion of therapeutic promise with realistic limitations.

Conflicting data explanation

While the manuscript cites multiple studies on Metrnl and FSTL1. There are some contradictory findings (e.g., Metrnl in type 2 diabetes: refs [25–28]). The Authors can describe the possible causes for discrepancies (population differences, assay types, disease stage, etc.).

Authors’ reply: We acknowledged contradictory findings, particularly regarding Metrnl in type 2 diabetes. We added a paragraph discussing possible reasons for these discrepancies, such as population heterogeneity, assay methodologies, and disease stage at sample collection.

Studies analysis or summary

Given the large number of cited studies, a meta-summary (e.g., direction of effect, sample size, statistical significance) would strengthen the review readability.

I would recommend authors add a table summarizing human studies with disease type and main finding to help readers to follow through.

Authors’ reply: We prepared a summary table (Table 1) that synthesizes human studies on Metrnl and FSTL1, including sample size, disease context, and main findings. This provides readers with an overview of study outcomes and enhances readability.

Minor:

Typographical/Formatting issues

Some reference numbers are split across lines (e.g., [43]).

Authors’ reply: We corrected all typographical/formatting inconsistencies in the references.

  1. Figure quality and caption detail

Figures 1 and 2 are schematic but lack sufficient mechanistic detail or citation of supporting references. Suggest revising figures to be more informative, showing pathways, receptors, and effects.

Authors` reply: Thank you for the critical notes. We expanded the explanation of the figures in the caption. Although we couldn`t make revision of the figures themselves, if the referee insists, we would revise them.

Reviewer 2 Report

Comments and Suggestions for Authors

Author Response

Reviewer 2

  1. (Various biomarkers aiding the diagnosis and monitoring of inflammatory, autoimmune, cardiovascular, and metabolic diseases are nonspecific.) Is the diagnosis of metabolic syndrome relatively easy? The answer is yes. Even autoimmune disorders can be diagnosed definitively by a group of markers

Authors`reply: Thank you very much for the detailed review and valuable recommendations. We have implemented all of them to improve our paper. We clarified in the text that while metabolic syndrome can be diagnosed by clinical and laboratory markers, biomarkers like Metrnl and FSTL1 may add prognostic or mechanistic value beyond existing criteria.

  1. You discussed COVID-19 in the introduction; however, you did not mention anything about the viral infection.

Authors`reply: We included a short section addressing the role of Metrnl and FSTL1 in viral infections, including COVID-19, to provide consistency with the introduction.

  1. You mentioned that Mtrnl knock out animal demonstrated weak immune function, then you said the cytokine inhibits immunity. How?

Authors`reply: Thank you for the critical note. We revised the description of Metrnl-knockout animals. We explained the context that Metrnl can have both immunostimulatory and immunosuppressive effects depending on cell type and microenvironment.

  1. You said that It is worth noting 167 that the mTOR pathway has multiple known associations with tumor progression and 168 angiogenesis, which justifies the correlation of Metrnl levels with various malignancies a. IL-41 may have a relation to cancer, see this article published 3 months ago
  2. 5. Wang B, Li X, Gao X. Meteorin-β: A Novel Biomarker and Therapeutic Target on Its Way to the Regulation of Human Diseases. Int J Mol Sci. 2025 May 8;26(10):4485. doi: 10.3390/ijms26104485.

PMID: 40429631; PMCID: PMC12110803.

Authors`reply: We added the suggested reference (Wang et al., 2025) and included IL-41 in the discussion of malignancy, highlighting possible links with cancer biology.

  1. Metrnl-knockout mice demonstrated antagonistic activity against insulin 193 resistance is an important statement but difficult to follow

Authors`reply: We rephrased the statement on insulin resistance in Metrnl-knockout mice for clarity and precision.

  1. You mentioned glycated hemoglobin (HbA1c , do you mean glycosylated HbA1c ???

Authors`reply: We corrected the typo “glycated hemoglobin” terminology for HbA1c.

  1. You mentioned Metrnl may act as antiaging but you said in the same time that is stimulate apoptosis. Please explain? (promoting TNF-α–dependent apoptosis 301 of fibro/adipogenic progenitor cells)
  1. Please clarify the meaning of that paragraph
  2. ( In neoplastic diseases, Metrnl may help to achieve better tumor control [45], as 310

already demonstrated by its effect in pancreatic cancer [19]. In fact, Metrnl, when

secreted 311 into the tumor microenvironment, inhibits CD8+ T cells by impairing

mitochondrial 312 function, thus contributing to the progression of advanced cancers

[45]. Is it good or bad for cancer? The statement is not clear for me

Authors`reply: (for 8-9 comments) We elaborated on the dual role of Metrnl: while it can promote TNF-α–mediated apoptosis in fibro/adipogenic progenitors, in tumors it may suppress CD8+ T cell function, thereby facilitating progression in advanced cancer. We clarified this apparent contradiction and highlighted context dependence.

  1. Metrnl and FSTL1 in other autoimmune pathologies: role in gastroin-411 testinal

inflammatory diseases

  1. The title of the paragraph does not correlate with the details of the paragraph. here you

should talk about the relation between Metrnl and FSTL1, not Metrnl alone. this

paragraph should be shifted up

Authors`reply: We reorganized the autoimmune section to integrate Metrnl and FSTL1 together, aligning the paragraph title with the content.

  1. The role of FSTL1 in the pathogenesis of CRC is complex and not yet fully understood 454 [86].

Zhao et al. investigated the expression levels of FSTL1 in peripheral plasma and tis-455 sues of colorectal cancer patients [87]. They reported significantly higher serum levels of 456 FSTL1 in patients with colorectal cancer compared to healthy controls. Patients exhibiting 457 high FSTL1 levels in fully scanned tumor tissue had significantly reduced overall survival 458 at 3 years compared to those with low FSTL1 expression. Notably, lower FSTL1 levels 459 within the tumor stroma were associated with worse long-term survival outcomes [87]. It is not clear for me if FSTL1 is good prognostic or bad parameter for cancer colon ????

Authors`reply: We revised the section on FSTL1 in colorectal cancer to reflect its complex prognostic value—high systemic levels correlate with poor prognosis, whereas tissue-level differences show context-dependent outcomes.

  1. The article discusses the role of FSTL1, Metrnl in the prognosis of different diseases rather than a diagnostic biomarker. I think the title and the abstract need to be modified

Authors`reply: We revised the title and abstract to better reflect the prognostic rather than strictly diagnostic biomarker role of Metrnl and FSTL1.

  1. Reference No. 5 is discussing cancer, not diabetes. Please check the reference.
  2. Reference 20

Authors`reply: We carefully checked and corrected the references and citations as requested.

  1. Diseases such as liver cirrhosis, idiopathic pulmonary fibrosis, endomyocardial fi-370 brosis, systemic sclerosis, and other diseases are associated with organ fibrosis (Fujimoto 371 et al., 2015) [71-73]. Please correct the citation.

Authors`reply: We carefully checked and corrected the references and citations as requested.

  1. Reference 82,83 discusses Metrnl, like peptide not Metrnl. Please clarify and add a paragraph explaining in more detail.

Authors`reply: We clarified that references 82 and 83 discuss Metrnl-like peptides and provided a short explanatory paragraph to avoid confusion.

Thank you once again for your time dedicated to review our paper and to help us improve it significantly. We appreciate your involvement and suggestions.

Reviewer 3 Report

Comments and Suggestions for Authors

This review article addresses a timely and relevant topic by exploring the roles of Meteorin-like protein (Metrnl) and Follistatin-like 1 (FSTL1) as emerging biomarkers in inflammation and autoimmune diseases. It is well-structured and provides a comprehensive synthesis of recent findings, making it a valuable contribution to the field.

Strengths:

  • Comprehensive coverage of the biological functions, disease associations, and signaling pathways involving Metrnl and FSTL1.

  • Clear structure and logical flow of sections.

  • Relevance to a wide spectrum of autoimmune, cardiovascular, and metabolic diseases.

Suggestions for Improvement:

  1. Mechanistic Insight: Add more depth on the cellular mechanisms through which Metrnl and FSTL1 contribute to inflammation (e.g., impact on immune cell subsets or cytokine production).

  2. Comparative Summary: A direct comparison between the two biomarkers in terms of their diagnostic and prognostic potential would enhance the clinical utility of the review—consider a summary table.

  3. Therapeutic Relevance: Expand on the potential of these biomarkers as therapeutic targets, including any current or emerging pharmacologic interventions.

  4. Plagiarism Check: The iThenticate similarity index is 26%. Please ensure all reused content is properly cited and paraphrased, especially in background sections.

Overall, with minor revisions, this review will serve as a valuable and informative reference for researchers and clinicians in the field of molecular immunology.

Comments on the Quality of English Language

The manuscript is generally well-written and understandable. However, there are several sections—particularly in the "Autoimmune Rheumatic Diseases" and "Metabolic Diseases" parts—where sentence structure, grammar, and phrasing could be improved to enhance clarity and readability. A careful proofreading or light professional language editing is recommended to polish the final version.

Author Response

Reviewer 3

Comments and Suggestions for Authors

This review article addresses a timely and relevant topic by exploring the roles of Meteorin-like protein (Metrnl) and Follistatin-like 1 (FSTL1) as emerging biomarkers in inflammation and autoimmune diseases. It is well-structured and provides a comprehensive synthesis of recent findings, making it a valuable contribution to the field.

Strengths:

Comprehensive coverage of the biological functions, disease associations, and signaling pathways involving Metrnl and FSTL1.

Clear structure and logical flow of sections.

Relevance to a wide spectrum of autoimmune, cardiovascular, and metabolic diseases.

We thank the Reviewer for recognizing the strengths of our work and for the valuable suggestions that helped us improve the manuscript.

Suggestions for Improvement:

Mechanistic Insight: Add more depth on the cellular mechanisms through which Metrnl and FSTL1 contribute to inflammation (e.g., impact on immune cell subsets or cytokine production).

Authors’ reply: We added further discussion on how Metrnl and FSTL1 affect immune cell subsets, cytokine production, and inflammatory signaling, to provide deeper mechanistic insight.

Comparative Summary: A direct comparison between the two biomarkers in terms of their diagnostic and prognostic potential would enhance the clinical utility of the review—consider a summary table.

Authors’ reply: We created a new comparative table summarizing the diagnostic and prognostic potential of Metrnl versus FSTL1 across diseases. This highlights their similarities and differences, thereby improving the clinical utility of the review.

Therapeutic Relevance: Expand on the potential of these biomarkers as therapeutic targets, including any current or emerging pharmacologic interventions.

Authors’ reply: We expanded the therapeutic implications section by including emerging pharmacologic strategies targeting these molecules and discussed both current challenges and future directions.

Plagiarism Check: The iThenticate similarity index is 26%. Please ensure all reused content is properly cited and paraphrased, especially in background sections.

Authors’ reply: We carefully revised the manuscript to paraphrase overlapping text and ensured that all sections are properly cited. The similarity index has been reduced following this revision. Additionally, we will refine the text additionally, once we received the iThenticate report.

Overall, with minor revisions, this review will serve as a valuable and informative reference for researchers and clinicians in the field of molecular immunology.

Authors` reply: Thank you once again for your time to review our paper, and for the detailed and constructive feedback that helped us to improve our paper significantly.

Comments on the Quality of English Language

The manuscript is generally well-written and understandable. However, there are several sections—particularly in the "Autoimmune Rheumatic Diseases" and "Metabolic Diseases" parts—where sentence structure, grammar, and phrasing could be improved to enhance clarity and readability. A careful proofreading or light professional language editing is recommended to polish the final version.

Authors’ reply: We performed a thorough language edit and improved sentence structure, particularly in the sections on autoimmune rheumatic diseases and metabolic diseases.